# A Subjective Logical Framework-Based Trust Model for Wormhole Attack Detection and Mitigation in Low-Power and Lossy (RPL) IoT-Networks

**Sarmad Javed [1], Ahthasham Sajid [1,2], Tayybah Kiren [3], Inam Ullah Khan [4], Christine Dewi [5,*], Francesco Cauteruccio [6,*] and Henoch Juli Christanto [7,*]**

1  Department of Computer Science, Faculty of Information and Communication Technology, Baluchistan University of Information Technology, Engineering and Management Sciences, Quetta 87300, Pakistan; ch.sarmad@hotmail.com (S.J.); gullje2008@hotmail.com (A.S.)
2  Faculty of Computing, Department of Computer Science, Capital University of Science and Technology, Islamabad 44000, Pakistan
3  Department of Computer Science, Rachna College of Engineering and Technology, University of Engineering and Technology, Lahore 54890, Pakistan; tayyaba@uet.edu.pk
4  Department of Electronic Engineering, School of Engineering & Applied Sciences (SEAS), Isra University, Islamabad 44000, Pakistan; inamullahkhan05@gmail.com
5  Department of Information Technology, Satya Wacana Christian University, Salatiga 50715, Indonesia
6  Department of Information Engineering, Polytechnic University of Marche, 60121 Ancona, Italy
7  Department of Information System, Atma Jaya Catholic University of Indonesia, Jakarta 12930, Indonesia
*  Correspondence: christine.dewi@uksw.edu (C.D.); f.cauteruccio@univpm.it (F.C.); henoch.christanto@atmajaya.ac.id (H.J.C.)

**Abstract:** The increasing use of wireless communication and IoT devices has raised concerns about security, particularly with regard to attacks on the Routing Protocol for Low-Power and Lossy Networks (RPL), such as the wormhole attack. In this study, the authors have used the trust concept called PCC-RPL (Parental Change Control RPL) over communicating nodes on IoT networks which prevents unsolicited parent changes by utilizing the trust concept. The aim of this study is to make the RPL protocol more secure by using a Subjective Logic Framework-based trust model to detect and mitigate a wormhole attack. The study evaluates the trust-based designed framework known as SLF-RPL (Subjective Logical Framework-Routing Protocol for Low-Power and Lossy Networks) over various key parameters, i.e., low energy consumption, packet loss ratio and attack detection rate. The achieved results were conducted using a Contiki OS-based Cooja Network simulator with 30, 60, and 90 nodes with respect to a 1:10 malicious node ratio and compared with the existing PCC-RPL protocol. The results show that the proposed SLF-RPL framework demonstrates higher efficiency (0.0504 J to 0.0728 J out of 1 J) than PCC-RPL (0.065 J to 0.0963 J out of 1 J) in terms of energy consumption at the node level, a decreased packet loss ratio of 16% at the node level, and an increased attack detection rate at network level from 0.42 to 0.55 in comparison with PCC-RPL.

**Keywords:** IoT; RPL; SLF; PCC

## 1. Introduction

The aim of this section is to introduce the background of IoT networks along with RPL protocol basics and the key challenges that Low-Power and Lossy Networks routing protocol face in general, focusing on the key challenge of security along with wormhole attacks to be detected and mitigated in this study.

The International Data Cooperation predicts that 75 billion remotes will connect to IoT devices by 2025, generating a large amount of sensitive data [1]. However, this increase in device usage also creates new vulnerabilities due to limited resources and mobility, making security a primary concern. The IoT and WSNs are becoming essential parts of

our lives, connecting devices via the internet to provide customization and aptitude in various fields. The IPv6 routing protocol for Low-Power and Lossy Networks (RPL) was formally introduced as the default routing protocol for the IoT in 2012 in an effort to connect and support IPv6 for IoT devices [2]. The RPL was developed with resource-constrained devices in mind, making it capable of handling a wide range of IoT-LLN applications. An active source and distance vector routing technique is the RPL. It creates Directed Acyclic Graphs (DAGs) to represent the network topology. A destination-oriented DAG (DODAG) is a DAG that is connected to a single root destination in the RPL. The latest technologies, 5G and 6G, incorporate advanced security techniques to support a vast infrastructure of interconnected IoT and WSN devices designed to move data over networks.

RPL is a routing protocol designed for Low-Power and Lossy Networks, which are typically used in IoT- and WSN-based devices to optimize traffic flow in network topology [3].

The study's primary focus on the wormhole attack within the RPL protocol highlights a foundational step in fortifying security for IoT and WSN devices. However, the limited scope may not fully encompass the diverse range of threats these networks face. Translating findings into practical implementation is hampered by hardware and energy constraints, while the dynamic nature of network topology poses adaptation challenges. As the model evolves to tackle multiple RPL attacks, scalability and trade-offs between security and efficiency must be carefully managed. Yet, assumptions, evolving attacks, and resource limitations may temper the model's real-world effectiveness. Navigating these complexities and aligning with regulatory and cost considerations will determine the model's overall impact on enhancing security for resource constrained IoT and WSN systems.

The security of IoT nodes, networks, and infrastructure is the most important problem. IoT devices communicate with one another using the routing protocol for Low-power and Lossy Networks (RPL). RPL has a lightweight core, so it does not allow methods for implementing security that require a lot of computation or resources. As a result, security attacks, which may be roughly divided into RPL-specific and sensor-network-inherited assaults, can target both the IoT and the RPL. Rank attacks and wormhole assaults, which are attack types inherited from sensor networks, are among the most worrisome protocol-specific attacks. By consuming, they aim for the RPL components and resources, such as control messages, repair processes, routing topologies, and sensor network resources.

The RPL protocol faces many key challenges, i.e., energy consumption, mobility, and density of nodes in terms of scalability. However, in this study, only the key issue of security has been addressed. However, the observation is that as the number of nodes increases in the network, the RPL protocol suffers in terms of quality services. The RPL protocol is commonly used for IoT network routing, but it has been found to be vulnerable to various attacks, including wormhole attacks [4]. However, RPL is vulnerable to multiple attacks due to the lack of a mechanism to monitor child behavior [5]. Three types of attacks against the RPL are the wastage of resources, the malicious alteration of RPL control messages, and the exploitation of network traffic transmission [6]. One of the most common attacks on the IoT and WSNs against the RPL is the wormhole attack, which changes the preferred parent of nodes and disrupts the DODAG tree topology [6]. These attacks can compromise the network and disrupt technology. PCC-RPL was used because it was state-of-the-art methodology at the time. The SLF-RPL is proposed to be an even better version of the PCC-RPL, considering the resource-constrained nature of the RPL in the IoT. That is what we have extracted from the PCC-RPL: if we move the security mechanism to another level of topology and specify the controller to make the calculations which were previously performed on the nodes, this would significantly improve the network lifetime, as proven by the derived results.

Researchers have proposed trust models to detect and mitigate these attacks, but many of these models are not efficient due to the resource constraints of IoT devices. To address this, a new trust model has been proposed that detects and mitigates wormhole attacks in an efficient manner while minimizing energy consumption and resource utilization. The

model has a separate controller layer to calculate trust values for nodes in the network. This study proposes a trust mechanism using a subjective logic framework (SLF) to detect and mitigate wormhole attacks on RPL-based IoT and WSN devices. The proposed model includes following research contributions.

- A secure version of RPL-based IoT and WSN devices to detect and mitigate wormhole attacks.
- A controller-based hierarchical model to calculate the trust metrics for each node separately utilizing the subjective logic framework.
- A system that enhances the QoS-based parameters, keeping in view the resource-constrained nature of the RPL-based IoT and WSN device.

This study aims to improve RPL security by detecting and mitigating wormhole attacks on RPL-based IoT and WSN devices while taking into account their constrained nature and resources. The study aims to maximize network life while improving security. The extensive use of IoT and WSN devices in daily life has led to an increase in security vulnerabilities. The limited power source, low memory, high heterogeneity, and limited processing power of these devices have led to security issues in existing routing protocols. Multiple strategies and methodologies for data transmission on the IoT and WSNs have been introduced in various studies. The layered topology of the IoT is described in Figure 1 previously.

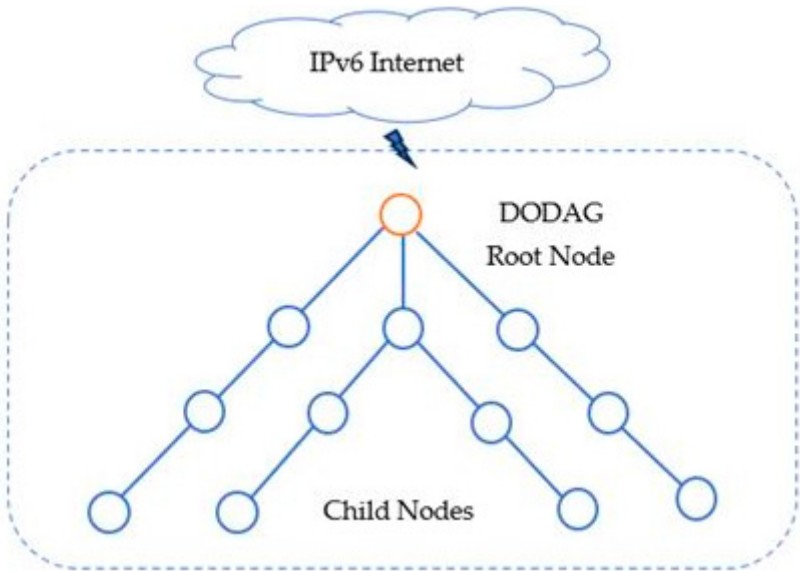

**Figure 1.** Topology of the IoT.

### 1.1. RPL

RPL is a custom-designed protocol for low-power IoT and WSN devices for efficient data transmission [7]. However, it is also vulnerable to attacks, which poses a challenge to securing data transmission while considering the resource-constrained nature of the nodes. In resource-constrained IoT devices, internal attacks are difficult to detect, and conventional security algorithms cannot be used due to limited resources [8]. This makes the timely detection and mitigation of attacks a challenging task. Therefore, new research is required to address these security issues.

### 1.2. RPL Security

Security concerns in the RPL pose challenges to its growth, especially in IoT and WSN applications, where data exchange between nodes follows a vulnerable route. Different attacks like Wormhole, Sybil, Fragmentation, Black hole, Rank, Clone ID, Version number, and Denial of Service attacks have made IoT nodes and network topologies vulnerable. Resource-constrained IoT nodes make it difficult to mitigate attacks, and different tech-

niques such as IDS, trust-based models, machine learning, and cryptography have been proposed to secure the RPL [9]. Trust-based models are easily adoptable and effective in detecting and mitigating attack nodes [10].

The proposed methodology in this study is trust-based to secure the RPL network. As described, RPL-based networks are vulnerable to multiple attacks. Figure 2 contains all the possible occurring attacks that are identified in the different studies.

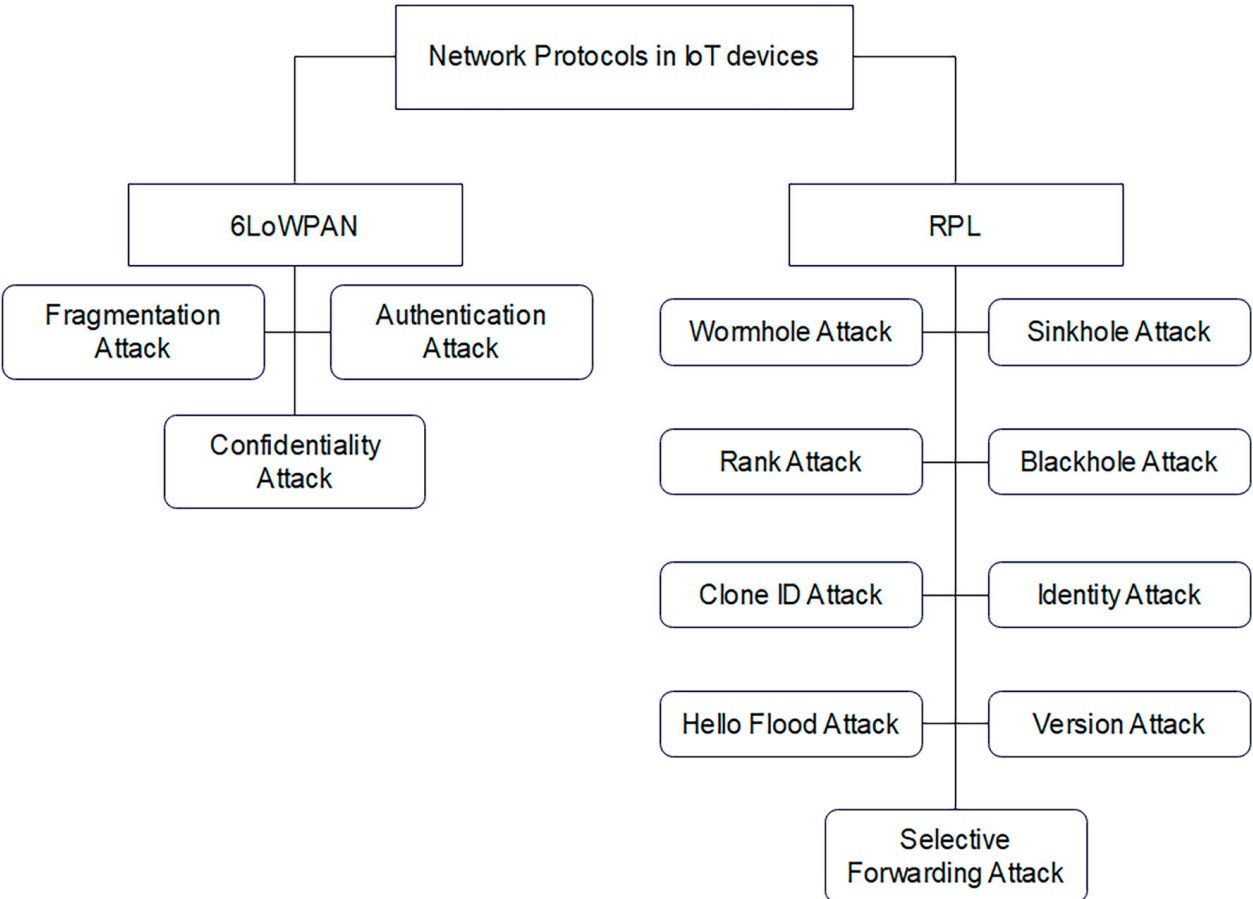

**Figure 2.** Classification of IoT Attacks over the RPL and 6LoWPAN.

*1.3. Wormhole Attack*

A wormhole attack is an internal attack on the RPL topology that disrupts the network by injecting a malicious node that trades off the integrity and confidentiality of the network topology, disrupts routing, and compromises nearby nodes [11]. The wormhole attack creates a tunnel between the source and destination nodes, enabling it to track and access confidential information. The attack can manipulate node behavior and pave the way for other unauthorized access. It makes two long-distance tunnels between nodes, and normal nodes get involved because of interference, resulting in transmission loss and power drain [12]. Wormhole attacks are hard to detect by localization protocols, and mitigation is challenging as the protocols take time to detect the attack [13].

Figure 3 describes how multiple wormhole attacks make a private tunnel and compromises the network topology.

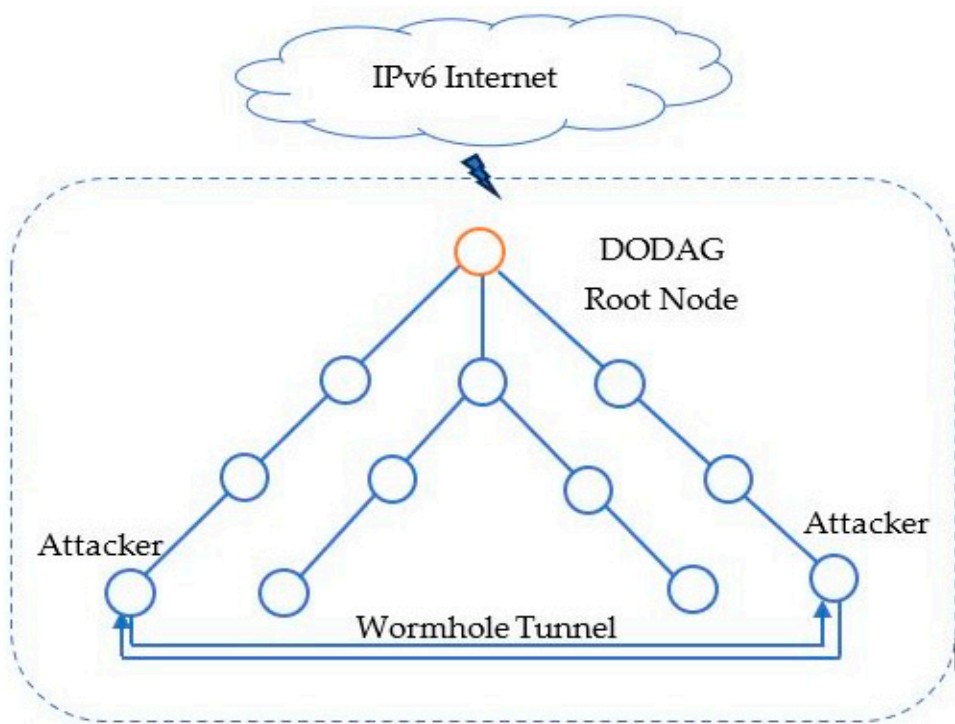

**Figure 3.** Wormhole Attack.

*1.4. SLF Trust Model*

The emergence of new research topics on trust models is due to the security challenges in IoT and WSN devices. Trust is defined as the effective communication between two nodes, which leads to secured data transmission and enhanced quality. This study proposes a trust model to mitigate wormhole attacks on WSN and IoT network topology using the Subjective Logic Framework (SLF) to evaluate trust from factors b, d, and u [14]. The recommendation of the node is checked, and the weight of their recommendation is calculated to ensure its accuracy. The proposed model is implemented and evaluated in Contiki's Cooja simulator using precision and recall methods.

The rest of the paper is organized as follows: In Section 2, existing hand-crafted and deep-learning feature-extraction and vehicle-classification methods are discussed briefly. In Section 3, the network architecture, along with pre-processing and dataset collection, are elaborated. The results and the comparison study are carried out in Section 4. Finally, the article conclusions are described in Section 5.

**2. Related Work**

In this section, the latest key research which has already been conducted to address the issue of routing will be critically reviewed and their findings analyzed and presented in a critical analysis table at the end of this section.

Several studies propose mechanisms to detect and prevent wormhole attacks, but most of them require excessive resource utilization at the node level. In one study, a new approach called the PCC-RPL (Parental Change Control RPL) was introduced to manage changes made by parents to IoT devices using the trust concept. This model's calculations are carried out at the node level of the topology [15]. In another study, a model was developed for enhancing RPL security and mitigating vulnerabilities to WSN-inherited and RPL-specific attacks. The researchers utilized machine-learning techniques, specifically supporting vector machines for data classification, to create a secure and improved version of the RPL for enhanced security [16]. One study identified parameters to monitor network flow and evaluation [17]. In another study, a mechanism model was proposed to detect and prevent wormhole attacks using the AODV protocol [18]. Another proposed a monitoring mechanism in which softwarized approach was used to detect

RPL based Attacks [19]. Another proposed a trust management framework that calculates and analyzes the trust of devices or nodes using MADM, and Evidence-Based Subjective Logic (EBSL) [20]. Another system is proposed as a secure version of the RPL that uses support vectors and a level-based approach to detect Sybil attacks [21]. Another proposed methodology for IoT security using leveraging blockchain technology [22]. However, our study proposed a "Subjective Logical Framework (SLF)-based Trust Model for Wormhole Attack Detection and Mitigation in IoT applications" to calculate trust values between trusted networks while saving resources at the node level. In another study to secure against Rank and Sybil attacks, the RPL routing protocol incorporated the Secure Trust (SecTrust) trust system. The SecTrust-RPL utilizes a trust-based mechanism to identify and contain attacks, all while optimizing network performance [23].

The study is analyzed using key parameters related to our study. The details are mentioned in Table 1 with their respective citations. Many of the cutting-edge mechanisms have often overlooked the resource limitations inherent in the RPL. Contemporary approaches have typically tasked ordinary nodes with implementing security measures, and while these methods have proven effective, they have also led to a significant drain on node resources. Therefore, this study aims to enhance and compare the results of the PCC-RPL methodology, which is the most recent technique to mitigate and detect wormhole attacks in RPL-based IoT devices. The PCC-RPL methodology worked well in terms of better network lifetime, attack detection rate, average residual energy, packet loss ratio, and attack detection time compared with other proposed methodologies. However, the node-level calculation and evaluation of trust metrics in the PCC-RPL resulted shorter network lifetime and high energy consumption by the nodes in comparison with SLF-RPL proposed method. However, a novel trusted model using routing can be utilized for attack detection [24].

**Table 1.** Critical Analysis Table.

| Citation | Year | Domain | Protocol | Summary of Related Work |
|---|---|---|---|---|
| SLF-RPL | | WSN; IoT | RPL | Presents a Subjective Logical Framework (SLF)-based Trust Model for Wormhole Attack Detection and Mitigation in WSN–IoT Applications. |
| [15] | 2021 | IoT | RPL | This study proposes the PCC-RPL (Parental Change Control RPL) methodology that controls the parent change mechanism of the IoT devices using the trust concept. The calculation in the model is performed at the node level of the topology. |
| [16] | 2021 | WSN/IoT | RPL | This study proposes a model for RPL security and addresses its vulnerabilities to both WSN-inherited and RPL-specific attacks. Study have used the machine-learning techniques to develop a secure and improved RPL version for the RPL security. The data classification is done in support vector machine for the analysis. |
| [18] | 2020 | WSN | AODV | This study proposes a model that uses the AODV protocol in its implementation and calculates the trust values of the nodes to detect and prevent a wormhole attack. |
| [21] | 2020 | WSN | LEACH | This paper proposes a method to mitigate the Sybil attack in the LEACH protocol-based WSN network. The study uses a three-tier detection designed for the attack in a severe environment. |
| [23] | 2017 | IoT | RPL | The Secure Trust (SecTrust) trust system is embedded into the RPL routing protocol to protect against Rank and Sybil attacks. The SunTrust-RPL uses a trust-based mechanism to detect and isolate attacks while optimizing network performance. |
| [25] | 2021 | IoT | NA | This study proposes a trust-based framework which computes the trust at the node level of the IoT devices. |
| [26] | 2020 | IoT | RPL | This study proposes the energy-based evaluation of the IoT devices. The evaluation is performed based on the reliability and energy efficiency of the devices. |

Therefore, this study proposes a methodology with a controller that calculates and evaluates results to cover up the gap. While the proposed methodology is based on device

layer, sink layer, controller layer, trust model, trust calculation, and trust model evaluation, the selection of nodes for routing and the isolation of the attacker node are discussed in detail.

In this section, the proposed SLF-RPL methodology is presented in detail along with the layer discussion, followed by a designed architectural diagram, flowchart, and trust calculation equations. In this study, the SLF-RPL hierarchical mechanism is proposed to detect and mitigate wormhole attacks in IoT applications using trust-based values. The study aims to improve the security of the RPL protocol, which is used for communication in IoT networks, by proposing an SLF-based trust model. The proposed solution is a three-layered architecture consisting of a device layer, a sink layer, and a control layer. Figure 4 explains all the functionality of the model in the form of an architectural diagram. The three-layered architectural diagram explains the work of the model.

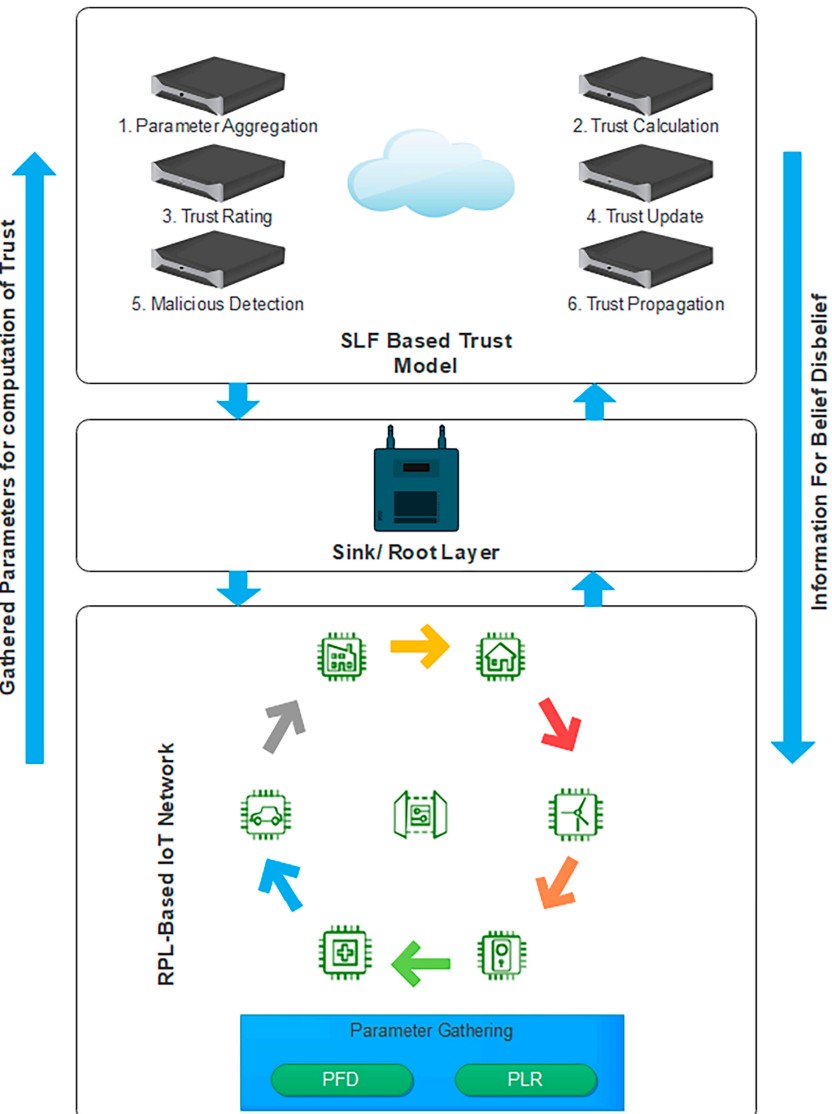

**Figure 4.** SLF-RPL Architectural Diagram.

## 3. Proposed Methodology (SLF-RPL)

### 3.1. Device Layer

The device layer is composed of IoT nodes such as home appliances and smart city devices, which are connected to the sink node or border router using the RPL protocol.

### 3.2. Sink Layer

The sink layer serves as the bridge between the controller and device layer.

### 3.3. Controller Layer

The controller layer is responsible for evaluating and computing collected values from the sink layer. It plays a crucial role in implementing the entire trust model process, including trust calculation, rating, evaluation, and updating the values linked to node-specific UID's. This layer enhances the capabilities of the device or nodes in the device layer, and it executes all operations.

In our methodology, a trust model managed by the SLF controller layer is further illustrated in Figure 4, and its details are mentioned below.

### 3.4. Trust Model

The study aims to minimize internal attacks using limited resources by minimizing trust-related data congestion and conserving energy consumption. To do this, a trust model is utilized, and the Subjective Logic Framework (SLF) is used for trust calculation. The trust model has a separate controller responsible for processing the parameters required for evaluation, which helps to save resources at the node level. The model gathers measurements of the required parameters from different child nodes in the RPL tree topology and transfers them to the controller for processing. The collected data is then used to calculate the trust for each node separately. The trust model helps nodes to identify malicious nodes with a wormhole attack. To manage the trust model, the SLF controller layer handles various tasks, as illustrated in Figure 5 below and further discussed in the following sections.

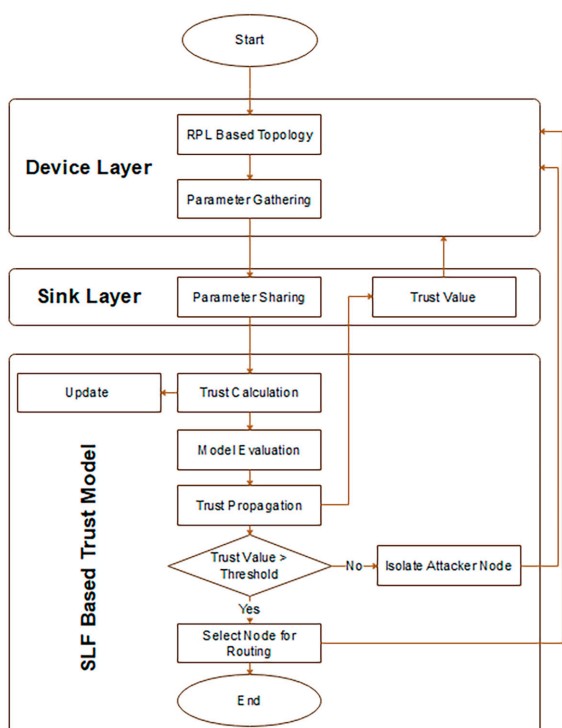

**Figure 5.** SLF-RPL Flow Chart.

### 3.5. Trust Calculation

The trust calculation in the Subjective Logic Framework (SLF) involves computing the trust value of nodes based on the belief, disbelief, and uncertainty parameters. The SLF considers the level of uncertainty in trust information and is built on the belief model. The trustworthiness of nodes is analyzed based on these parameters, and the opinion is based on belief (B), disbelief (D), uncertainty (U), and the value of α. The parameter of

belief assigns nodes as trusted; the disbelief parameter isolates the node from the topology, and uncertainty marks the node as unknown. The value of $\alpha$ represents the uncertainty result of beliefs and disbelief, and the sum of B, D, and U equals 1. The previous results of the trustworthiness of nodes are stored based on whether the node was pre-trusted or not, which is important when registering new nodes.

The value of $\alpha$ depends on whether the node was pre-trusted or not. This is important when registering the new nodes. The value is calculated as

$$\alpha = \frac{1.0 \text{ (if node pre trusted)}}{0.5 \text{ otherwise}} \tag{1}$$

To calculate the trust for object I, Trust of object "I" is calculated as

$$w_{i=(b_i,d_i,u_i)} \tag{2}$$

The relationship between belief, disbelief, and uncertainty is described by the below Opinion Triangle in Figure 6. When the value of uncertainty is low, it will be at the bottom of the triangle. When value of uncertainty is not low, the node has abundant experience the uncertainty level will be high.

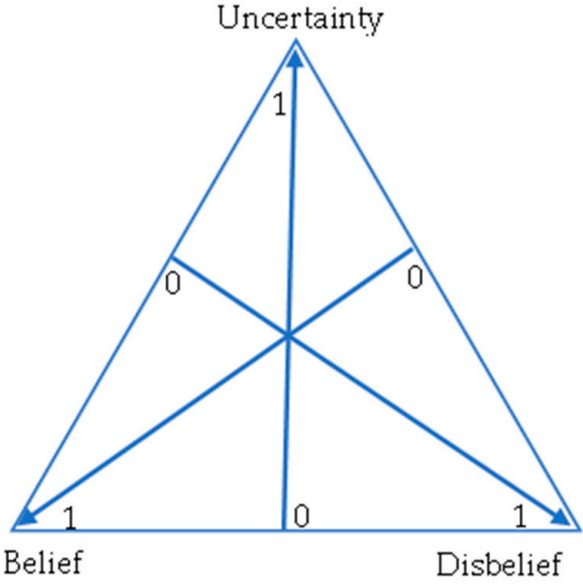

**Figure 6.** Opinion Triangle.

### 3.6. Trust Aggregation

The subjective logic framework ensures that the network topology will have trusted nodes for data sharing. It checks the self-experience of the devices, considering whether the parameters b, d, and u are trusted or not. The process is called recommendation of the node. To ensure that their recommendation is true, it is necessary that calculation of the recommendation value is performed. To calculate the global trust value, the SLF evaluates all the gathered values of each node to calculate the recommendation values. To calculate the recommendation value, two methods are used, namely the discounting and the consensus methods.

### 3.7. Discounting Method

The discounting operator denotes the discounting method as $\otimes$ [27]. For the communication of nodes A to C, the discounting method takes recommendations from the B node. Total trust on the C node is the combination of trust B for C and trust A for B, which is illustrated in Figure 7 below.

$$w_b^a = (b_b^a,\ d_b^a,\ u_b^a) \tag{3}$$

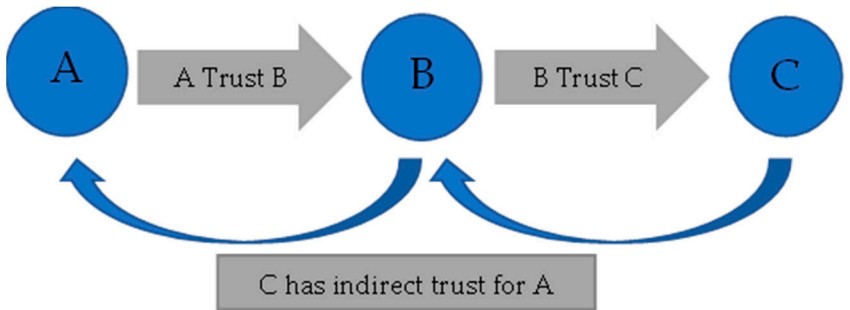

**Figure 7.** Node Trust.

The trust that B has for C is

$$w_c^b = \left(b_c^b,\ d_c^b,\ u_c^b\right) \tag{4}$$

Therefore, the trust that A has for C is

$$w_c^a = w_b^a \otimes w_c^b = \left(b_b^a, b_c^b\right),\ \left(b_b^a,\ d_c^b\right),\ \left(d_b^a + u_b^a + b_b^a u_c^b\right) \tag{5}$$

*3.8. Trust Model Evaluation*

Trust evaluation is a crucial step in the proposed model, as it helps to establish the initial trust between nodes and dynamically updates it with measured parameters as needed. The evaluation process compares the calculated trust values with a threshold value for each node in order to establish trust relations between nodes and their parent nodes for secure communication within the network topology. The evaluation process in the Subjective Logic Framework considers belief, disbelief, and uncertainty, with a threshold value of 0.5 for all parameters as presented in Table 2 below. The controller stores the trust-related measurements with a unique identifier (UID) for each device generated through a random function and evaluates trust values separately against the UIDs of each device.

**Table 2.** Node Routing.

| Belief (b) | Disbelief (d) | Uncertainty (u) | Rating |
| --- | --- | --- | --- |
| If b > 0.5 | If d < 0.5 | If u < 0.5 | Node is Trusted |
| If b < 0.5 | If d > 0.5 | If u < 0.5 | Node is not Trusted |
| If b < 0.5 | If d < 0.5 | If u > 0.5 | Node is not Verified |
| If b $\leq$ 0.5 | If d $\leq$ 0.5 | If u $\leq$ 0.5 | Node is not Verified Yet |

*3.9. Selecting Nodes for Routing*

In this phase of the trust-based routing model, the evaluated trust values of the nodes are used to establish a trusted network. The nodes with trust values greater than the set threshold value are selected for routing, and a trusted route is created between the root and child nodes. This ensures that only nodes with a certain level of trustworthiness are used for communication, thus reducing the risk of data being compromised or intercepted by malicious nodes. The trusted route serves as a secured channel for the nodes to exchange data between the trusted networks.

*3.10. Isolating the Attacker Node*

In this phase, the trust values calculated in the previous phases are used to determine if a node is trustworthy or not. If the calculated trust value of a node is less than the set threshold value, the model isolates the node from the trusted network, as it is considered malicious. This helps to prevent further attacks and ensures the security of the network.

## 4. Experiments and Results

This section describes the evaluation and results of a mechanism designed for securing IoT networks. The simulation is carried out using the Cooja simulator 3.0; the simulation setup details in Table 2 are presented in Section 4.1, experimentation details are provided in Section 4.2, and the results and discussion are presented in Section 4.3.

The proposed solution of the PCC-RPL is found to be superior in terms of power consumption, packet loss ratio, and the number of detected attacks. The SLF-RPL mechanism operates in the No Horizontal position mode and forwards packets to the sink node in the No Down mode. The reception ratio ranges from 30% to 100%. The simulations were carried out in three different scenarios, with 33, 66, and 99 nodes, and a 65 m interference limit was established. The Contiki Cooja simulator was chosen because it is specifically designed for IoT and WSN networks, while other simulators are designed for large infrastructure networks.

### 4.1. Simulation Setup

Figure 8 shows a snapshot of the Contiki Cooja simulator and a suggested scheme architecture for the first scenario (33 nodes), where node 33 is the control. Nodes 28 through 30 are malicious nodes, nodes 31 and 32 are sink nodes, and the number of nodes is typical. Table 3 explains the simulation environment set for the implementation.

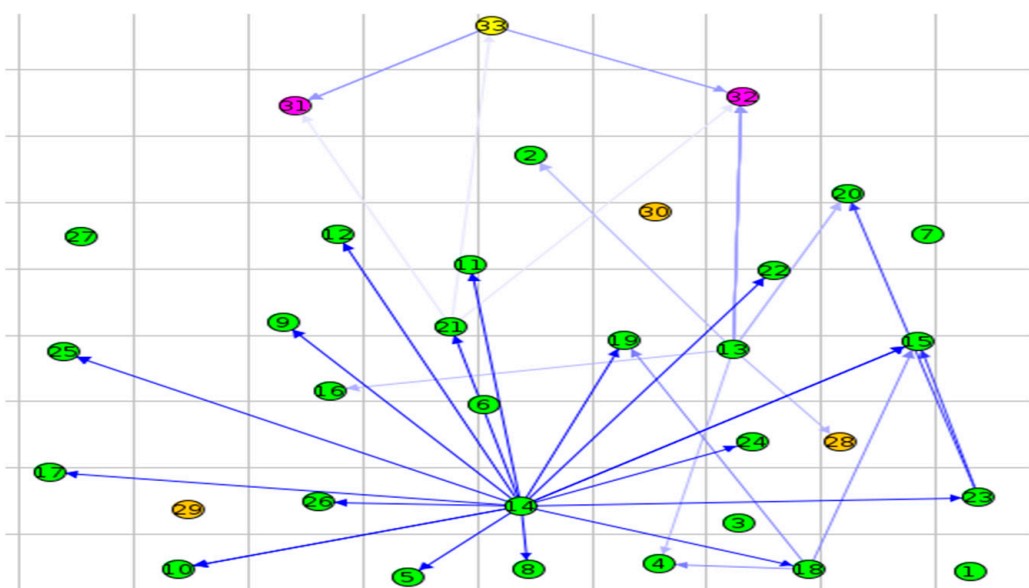

**Figure 8.** Simulation Environment.

The simulations are performed using the following resources. Hardware resource: Core i7 processor; RAM: 8 GB; software resource: VMware Workstation 16 player is used to run the Contiki OS. We have used the Cooja 3.0 simulator for the implementation and evaluation of our trust model.

**Table 3.** Simulation Environment.

| Simulation Parameters | Value |
|---|---|
| Simulation Tool | Contiki/Cooja 3.2 |
| Routing Protocol | RPL |
| Coverage Area | 100 m × 100 m |
| Simulation Time | 30, 60 and 90 min |
| Simulation Scenarios | 03 |
| # of Nodes in Topology | 30, 60, 90 |
| Attacker Nodes with Ratio | 3, 6, and 9 |
| Legitimate vs. Malicious Nodes Ratio | 1:10 |
| RX Ratio | 40% to 100% |
| TX Ratio | 100% |
| TX Range 50 m | 60 m |
| Inference Range | 65 m |
| Initial Node Energy | 0.1 J |

### 4.2. Experimentation

In this study, three experiments were performed with different numbers of nodes and time. Table 4 shows the different experiments performed with the number of deployed nodes, types of nodes, and runtime in each experiment.

**Table 4.** Experiments.

| Experiments | Deployed Nodes | Attacker Nodes | Controller Nodes | Border Router | Runtime in Minutes |
|---|---|---|---|---|---|
| Ex 1 | 30 | 3 | 1 | 1 | 30 |
| Ex 2 | 60 | 6 | 1 | 2 | 60 |
| Ex 3 | 90 | 9 | 1 | 3 | 90 |

### 4.3. Results Analysis

In this section, the implementation and evaluation of a wormhole attack are described, along with the identification of the attack, the attack detection rate, Packet loss ratio, energy consumption, average residual energy, and attack detection time. The proposed method is compared against the PCC-RPL mechanism, which uses an MRHOF optimization problem to choose routes. The comparison is based on the packet loss ratio and energy usage, as MRHOF does not offer wormhole discovery. The evaluation results show improved performance of the proposed method in terms of the selected parameters.

#### 4.3.1. Wormhole Implementation

Wormhole initially acts in a trustworthy manner for the first five to ten seconds during an experiment. However, after this period, it starts to behave maliciously by discarding data packets received from nearby nodes instead of forwarding them to their intended destinations. As a result, the RPL-based network experiences significant degradation because the nodes continue to transmit data packets to the malicious wormhole node.

#### 4.3.2. Attack Detection Rate

In this section, the effectiveness of the proposed solution for detecting and isolating wormhole threats during routing is discussed. Figure 9 illustrates the comparison between the proposed SLF-RPL mechanism and the PCC-RPL. The SLF-RPL was found to have a higher detection ratio compared to the PCC-RPL. Initially, both systems detected multiple harmful nodes, but as the trust became fully implemented, the number of malicious nodes significantly decreased. At 10 min, the detection rate for the SLF-RPL was 0.56, while for the PCC-RPL, it was 0.45. At time thirty, the detection rates for the SLF-RPL and the

PCC-RPL were 0.78 and 0.65, respectively. During the simulation, SLF-RPL detected all the malicious nodes, while the PCC-RPL remained unsuccessful in detection.

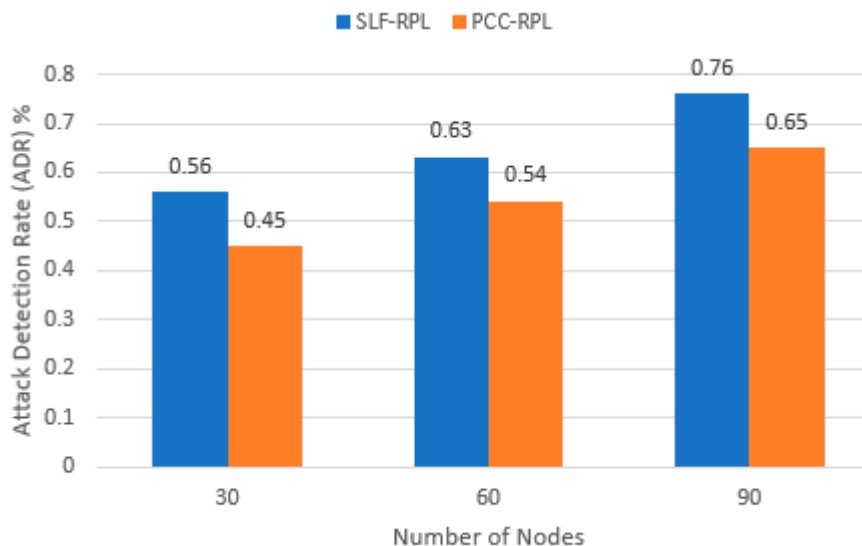

**Figure 9.** Attack Detection Rate at the Network Level.

Furthermore, the average detection rate for different numbers of nodes (30, 60, and 90) is shown in Figure 10, demonstrating the effectiveness of the proposed solution (SLF-RPL).

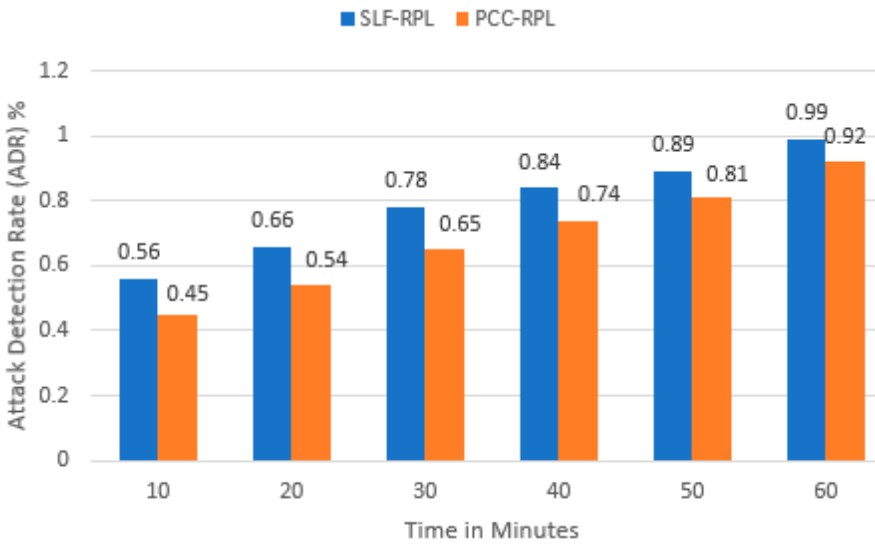

**Figure 10.** Attack Detection Rate at the Node Level.

Overall, the proposed solution showed promising results in detecting and isolating wormhole threats during routing.

### 4.3.3. Packet Loss Ratio

In wireless sensor networks, the packet loss ratio is an important metric that measures the number of received packets in relation to the total number of packets transmitted. Our proposed SLF-RPL mechanism effectively detects and mitigates malicious nodes, thereby reducing unnecessary overheads on the DODAG caused by wormhole packet loss. In comparison to the PCC-RPL mechanism, our SLF-RPL mechanism demonstrates superior performance in terms of the packet loss rate, as evidenced by Figure 11. Our mechanism exhibits a 16% lower packet loss rate than the PCC-RPL, with an average packet loss rate of 0.33 as opposed to the PCC-RPL's 0.38.

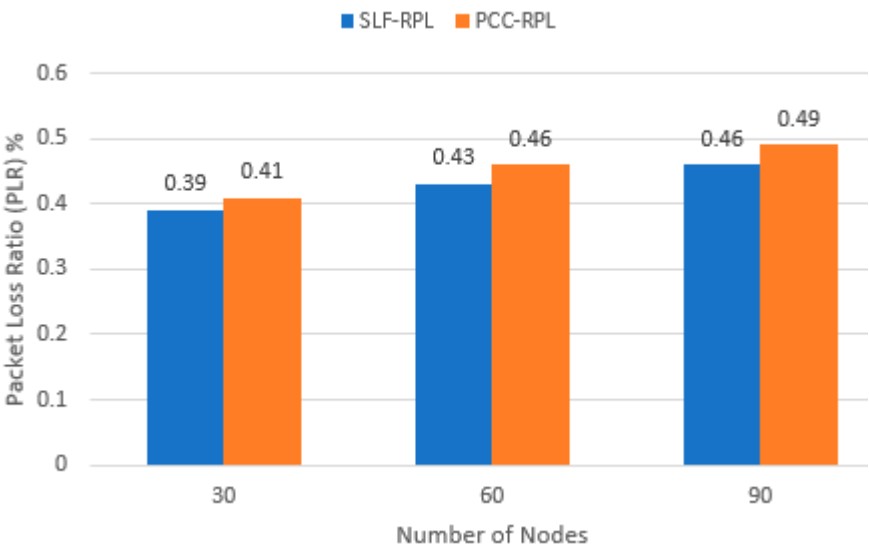

**Figure 11.** Packet Loss Ratio at the Node Level.

Average packet loss rates for different numbers of nodes (30, 60, and 90) are shown in Figure 12.

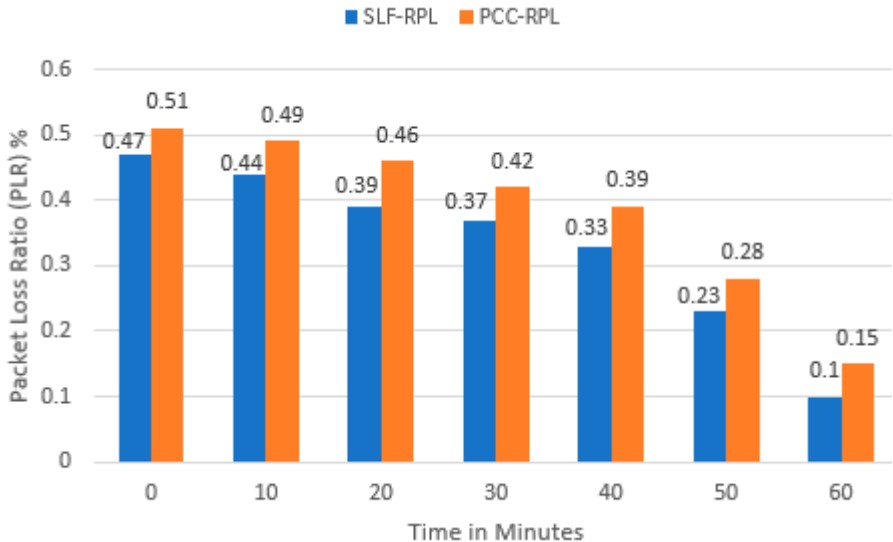

**Figure 12.** Average Packet Loss Ratio at the Network Level.

Consequently, our approach offers a better defense against wormhole attacks and a lower packet loss rate.

### 4.3.4. Energy Consumption

In this study, a trust method is proposed for Low-Power and Lossy Networks (LLNs), which are found to be 30% more energy efficient than the PCC-RPL, as illustrated in Figure 13. The average energy consumption (J) used by our mechanism is 0.042, while the PCC-RPL consumed 0.057.

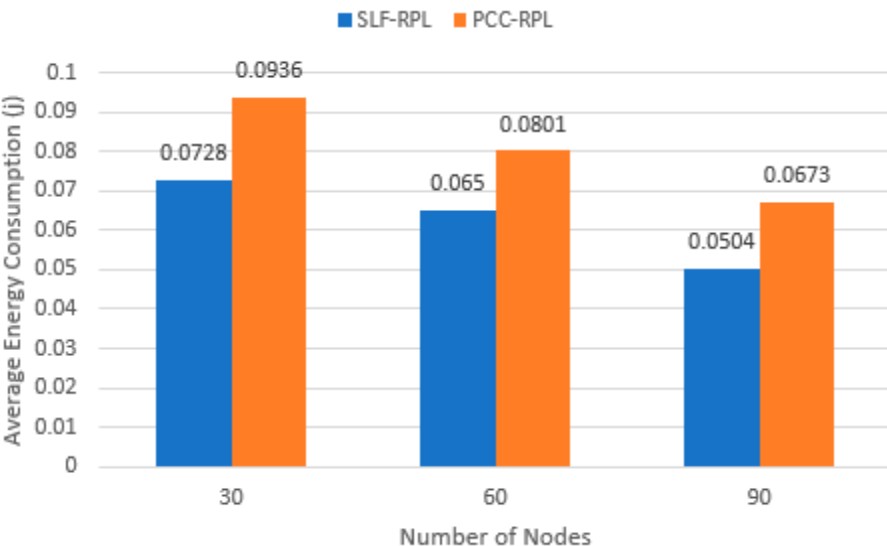

**Figure 13.** Energy Consumption at the Node Level.

The PCC-RPL uses less energy than some other OFs, but it does not have any attack mitigation mechanism to manage additional packets dropped by malicious nodes in the network. On the other hand, our mechanism detects and mitigates such nodes, resulting in lower energy consumption. Despite having a system to control fraudulent nodes, the PCC-RPL still uses more power than our method does, as illustrated in Figure 14.

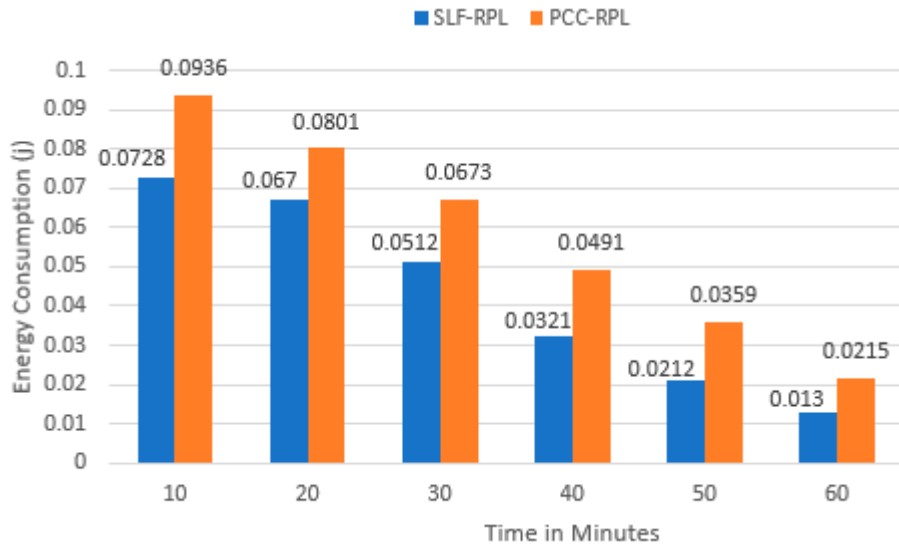

**Figure 14.** Average Energy Consumption at the Network Level.

4.3.5. Average Residual Energy

In the study, Figure 15 shows the average remaining energy of nodes after the simulation. The findings indicate that our proposed method, compared with the PCC-RPL, can save 8% more of the remaining power for nodes. The chart displays the aggregate residual energy for the nodes at different times during the simulation, starting at one mJ at one time. The nodes' energy gradually drains out over time, and the PCC-RPL shows a significant energy loss at time twenty-nine, which remains slow for the rest of the simulation. Our mechanism has more remaining energy than the PCC-RPL because our trust model's processes operate on the controller in the control plane rather than on IoT networks, reducing the energy overhead at the deployed nodes. The nodes on the device layer only receive data about their neighboring nodes and forward it to the control plane through a sink node.

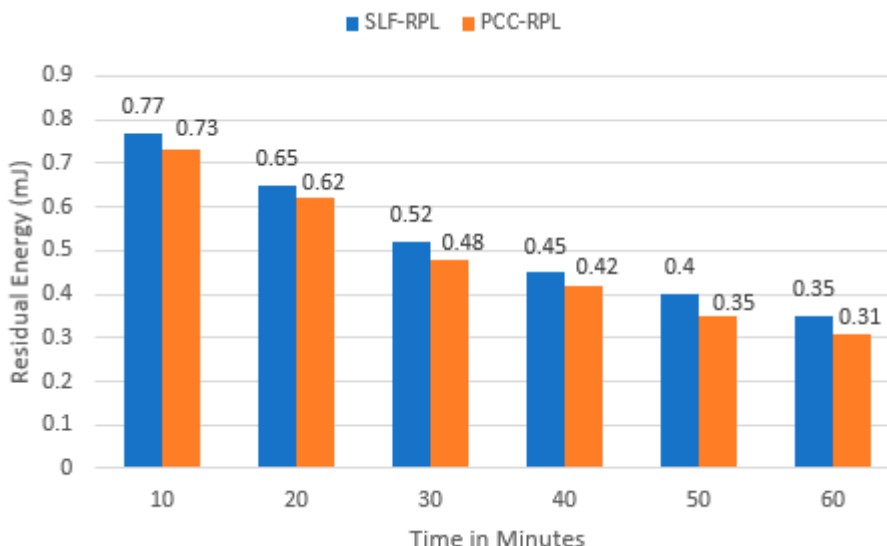

**Figure 15.** Residual Energy at the Network Level.

Figure 16 represents the average residual energies at different numbers of nodes. Average residual energies are calculated on 30, 60, and 90 number of nodes respectively for a better evaluation of our proposed solution.

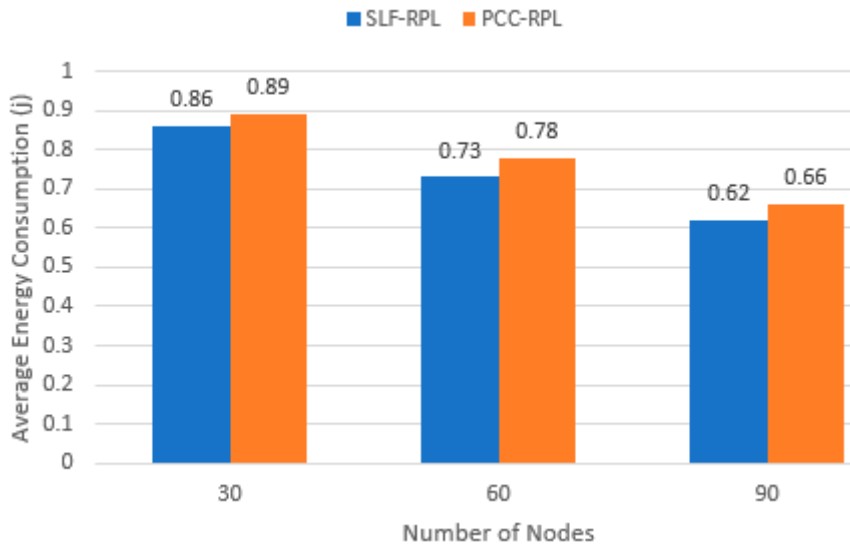

**Figure 16.** Average Residual Energy at the Node Level.

4.3.6. Attack Detection Time

Attack detection time refers to the time taken by the controller to identify if there is an attacker node present in the network, and it is calculated by subtracting the time for traffic to start from the time it takes to detect an attack. In this study, the proposed solution, SLF-RPL, is compared with the PCC-RPL in terms of attack detection time. The results showed that the SLF-RPL outperformed the PCC-RPL due to its veracity, and the graph in Figure 15 illustrates that the SLF-RPL had a lower attack detection time, resulting in a longer network lifetime as the number of nodes increased, as illustrated in Figure 17.

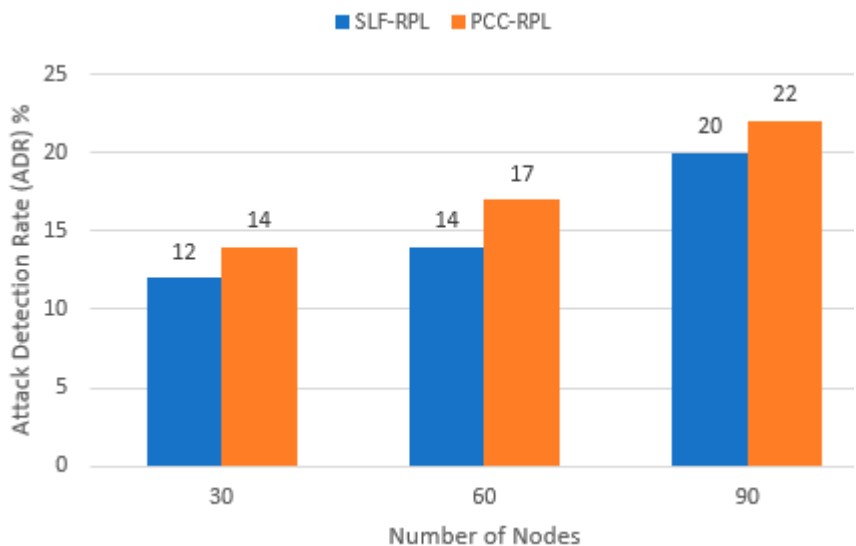

**Figure 17.** Attack Detection Time at the Node Level.

### 4.4. Discussion of Results

The provided data outlines the key evaluation parameters for various simulation scenarios involving different quantities of nodes (30, 60, and 90) using two distinct routing protocols: PCC-RPL and SLF-RPL. The metrics examined encompass the Packet Loss Ratio (%), Energy Consumption (measured in Joules), and Attack Detection Rate (expressed in minutes).

Upon analyzing the data, it becomes apparent that the two routing protocols, PCC-RPL and SLF-RPL, showcase varying performance characteristics across different scenarios.

In terms of the Packet Loss Ratio, the SLF-RPL consistently demonstrates marginally superior results compared with the PCC-RPL. The recorded values range from 0.39% to 0.46% for the SLF-RPL and from 0.41% to 0.49% for the PCC-RPL. This indicates that the SLF-RPL tends to maintain a slightly lower packet loss ratio, suggesting potentially more reliable data transmission, as presented in Table 5 below.

**Table 5.** Results Comparison of PCC-RPL vs. SLF-RPL at the Node Level.

| Key Evaluation Parameters | Simulation Scenario: 30 Nodes | | Simulation Scenario: 60 Nodes | | Simulation Scenario: 90 Nodes | |
|---|---|---|---|---|---|---|
| | PCC-RPL | SLF-RPL | PCC-RPL | SLF-RPL | PCC-RPL | SLF-RPL |
| Packet Loss Ratio (%) | 0.41 | 0.39 | 0.46 | 0.43 | 0.49 | 0.46 |
| Energy Consumption (J) | 0.0963 | 0.0728 | 0.0801 | 0.065 | 0.0673 | 0.0504 |
| Attack Detection Rate (Time in Mints) | 12 | 14 | 14 | 17 | 20 | 22 |

As for Energy Consumption, both protocols exhibit an overall reduction in energy usage as the number of nodes increases. Notably, the SLF-RPL consistently achieves better energy efficiency across all scenarios. The energy consumption figures fluctuate between 0.0504 J and 0.0728 J for the SLF-RPL, while the PCC-RPL's values range from 0.065 J to 0.0963 J, as presented in Table 5. This indicates that the SLF-RPL is more adept at conserving energy, which is crucial for prolonging the network's operational lifespan.

Turning to the Attack Detection Rate, it is observed that the SLF-RPL demonstrates faster attack detection times across all simulation scenarios. The detection times range from 12 to 22 min for the SLF-RPL, while the PCC-RPL achieves detection in a slightly

broader time frame, ranging from 14 to 20 min, as presented in Table 5. This suggests that the SLF-RPL is more responsive in identifying potential attacks within the network.

In summary, the data present evaluations of different simulation scenarios involving node quantities of 30, 60, and 90, utilizing two routing protocols, PCC-RPL and SLF-RPL. The analyzed metrics include Packet Loss Ratio, Energy Consumption, and Attack Detection Rate. The SLF-RPL consistently exhibits slightly better Packet Loss Ratio results (0.39% to 0.46%) compared with the PCC-RPL (0.41% to 0.49%), implying more reliable data transmission. In terms of Energy Consumption, the SLF-RPL demonstrates higher efficiency (0.0504 J to 0.0728 J) than the PCC-RPL (0.065 J to 0.0963 J), indicating better energy conservation. Notably, the SLF-RPL also excels in the Attack Detection Rate, achieving faster detection times (12 to 22 min) compared with the PCC-RPL (14 to 20 min), showcasing a heightened responsiveness to potential network attacks, as presented in Table 5.

Table 6 provides a comprehensive and detailed analysis of key evaluation parameters for the two distinct routing protocols, PCC-RPL and SLF-RPL, across multiple time intervals ranging from 10 to 60 min. These parameters, namely Attack Detection Rate (%), Average Residual Energy (J), and Packet Loss Ratio (%), shed light on the performance and efficiency of the two protocols under different conditions.

**Table 6.** Results Comparison of PCC-RPL vs. SLF-RPL at the Network Level.

| Key Evaluation Parameters | 10 Min | | 20 Min | | 30 Min | | 40 Min | | 50 Min | | 60 Min | |
|---|---|---|---|---|---|---|---|---|---|---|---|---|
| | PCC-RPL | SLF-RPL | PCC-RPL | SLF-RPL | PCC-RPL | SLF-RPL | PCC-RPL | SLF-RPL | PCC-RPL | SLF-RPL | PCC-RPL | SLF-RPL |
| Attack Detection Rate (%) | 0.45 | 0.56 | 0.54 | 0.66 | 0.65 | 0.78 | 0.74 | 0.84 | 0.81 | 0.89 | 0.92 | 0.99 |
| Average Residual Energy (J) | 0.73 | 0.77 | 0.62 | 0.65 | 0.48 | 0.52 | 0.42 | 0.45 | 0.35 | 0.40 | 0.31 | 0.35 |
| Packet Loss Ratio (%) | 0.51 | 0.47 | 0.49 | 0.44 | 0.46 | 0.39 | 0.39 | 0.33 | 0.28 | 0.23 | 0.15 | 0.1 |

Starting with the Attack Detection Rate (%), which signifies the protocols' ability to identify and respond to network attacks, the SLF-RPL consistently outperforms the PCC-RPL across all time intervals. This suggests that the SLF-RPL possesses a more robust and responsive mechanism for detecting potential security threats within the network. The increasing trend in both protocols' attack detection rates as the time interval grows reflects their improving adaptability over longer monitoring periods.

Moving on to the Average Residual Energy (J), a critical metric indicating the energy consumption efficiency of the routing protocols, the SLF-RPL maintains a notable advantage over the PCC-RPL. The decreasing values of average residual energy for both protocols over the time intervals indicate a potential energy conservation effect. However, the SLF-RPL consistently maintains lower energy consumption levels compared with the PCC-RPL, indicating its proficiency in optimizing energy usage, which is particularly advantageous for prolonging network lifetime and supporting resource-constrained devices.

The Packet Loss Ratio (%) reveals the reliability of data transmission within the network. The SLF-RPL consistently exhibits lower packet loss ratios compared with the PCC-RPL across various time intervals. This suggests that the SLF-RPL offers enhanced data transmission stability and integrity, which is crucial for maintaining a seamless communication environment. The diminishing trend in packet loss ratios for both protocols as the time interval increases indicates their improved data transmission reliability over extended periods.

In summary, Tables 5 and 6 detail evaluation parameters that provide a comprehensive insight into the comparative performance of the PCC-RPL and SLF-RPL routing protocols. The SLF-RPL demonstrates superiority in terms of attack detection rate, energy efficiency, and packet loss ratio across all evaluated time intervals. These findings underscore the potential benefits of SLF-RPL in enhancing network security, optimizing energy consumption, and ensuring reliable data transmission, making it a promising candidate for various real-world applications.

## 5. Conclusions

This study addresses the security issues in the RPL protocol for IoT and WSN devices, focusing on wormhole attacks, which are serious threats to the network. The research proposes a separate controller for the trust calculation to minimize energy consumption and packet loss and to increase the network lifetime. The model is evaluated using different parameters such as network lifetime, packet loss ratio, energy consumption, average residual energy, and detection time. The proposed method uses a Subjective Logic Framework-based control, resulting in a decrease in energy consumption, a 10% decrease in packet loss ratio, and an increase in attack detection time in comparison with the PCC-RPL. The study concludes that this model can effectively detect and mitigate wormhole attacks on RPL-based IoT and WSN devices, making it a promising approach for future research. The result of the study will enable researchers to continue working out the betterment of security for resource-constrained devices of the IoT and WSNs using RPLs. This model can be extended to tackle other attacks on the RPL protocol.

The study successfully surmounted numerous challenges by adopting a distinct topology in contrast to existing methodologies. Notably, the SLF-RPL introduces an additional layer of security beyond the node level for trust calculations, yielding enhanced resource utilization. While it demonstrated effectiveness in simulations, translating this success to real-time applications may pose deployment challenges.

The study exhibits certain limitations. Firstly, although the proposed SLF-RPL framework demonstrated efficacy in mitigating wormhole attacks, its adaptability for addressing various other low-power device vulnerabilities within the IoT landscape remains unexplored. This presents an avenue for future research to extend its applicability to a broader spectrum of security challenges.

Secondly, the study's scope is confined to the RPL protocol in the IoT domain. Nevertheless, there exists potential for enhancing its performance by extending its applicability to diverse IoT protocols through the integration of advanced artificial intelligence and deep-learning techniques. Techniques such as machine and deep learning models will be used to detect and isolate internal attacks as suggested by authors in [28] ant colony optimization, Markov chain distribution, and the deep neural networks described in can be tested within the framework of the SLF-RPL to further optimize attack detection rates, thereby expanding its utility and impact in safeguarding IoT ecosystems.

**Author Contributions:** Conceptualization, S.J., A.S. and T.K.; methodology, S.J., A.S. and T.K.; software, I.U.K., A.S. and S.J.; validation, S.J. and A.S.; formal analysis, T.K.; investigation, I.U.K.; resources, writing—original draft preparation, I.U.K. and C.D.; writing—review and editing, F.C., C.D. and H.J.C.; visualization, H.J.C. and F.C.; supervision, A.S. and I.U.K.; project administration, C.D. and F.C.; funding acquisition, F.C. and H.J.C. All authors have read and agreed to the published version of the manuscript.

**Funding:** This research received no external funding.

**Institutional Review Board Statement:** Not applicable.

**Informed Consent Statement:** Not applicable.

**Data Availability Statement:** The data are currently not publicly available due to participant privacy, but they are available from the corresponding author upon reasonable request.

**Acknowledgments:** All the authors would like to give thanks for the support and help from Satya Wacana Christian University and to others who took part in this work.

**Conflicts of Interest:** The authors declare no conflict of interest.

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
