# Peer review of "A Subjective Logical Framework-Based Trust Model for Wormhole Attack Detection and Mitigation in Low-Power and Lossy (RPL) IoT-Networks"

_information, doi:10.3390/info14090478_

Round 1
Reviewer 1 Report
Enclosed please find the attachment.

Author Response
Dear Reviewer,
We thank for your valuable comments. Please see the attachment reporting the point-by-point response.

Reviewer 2 Report
In the introduction part of the paper authors claim that "Results showed that the SLF-RPL model had 30% less energy consumption, decreased packet loss ratio by 16%, and increased attack detection rate from 0.42 to 0.55 compared to PCC-RPL." Authors must take into account the energy consuption of both models (if not already made) and clearly explain (prove with data included for energy consumption of both models) the calculation (algorithm).
Author Response

(The authors gave the same response as above.)

Reviewer 3 Report
Dear Author,
This paper needs a lot of efforts to put on. Please see my comments:
1. I would strongly suggest to focused on researched writing; it is different than academic writing. Introduction should be in the form of background and rationale, key challenges, objectives, key takeaways from your work, implications in short and outline. You need to rewrite the Introduction entirely.
2. Abstract - Please use figures to outline your research work. Implication required in 1-2 lines at the end.
3. Whenever you write a new section, please use objective or purpose of that section as a first para. for example, background section.
4. It would be great if you can give of your research work and refer that name throughout paper.
5. All Figures must be redrawn with Graphical tools
6. My big concern is the "discussion" part. I couldn't see any discussion. In discussion, what I expect to see is that what are the plausible reasons behind obtaining these results? Moreover, the validity and accuracy of the data should be discussed. The authors must comment on whether or not the results were expected and present explanations for the results; they must go into greater depth when explaining findings that were unexpected or especially profound. If appropriate, note any unusual or unanticipated patterns or trends that emerged from your results and explain their meaning. Be sure to advocate for your findings and underline how your results significantly in move the field forward. Remember to make sure you give your results their due and not undermine them. Moreover, in discussion, you should clearly state what your study adds to the body of the literature.
Must be improved.
Author Response

(The authors gave the same response as above.)

Round 2
Reviewer 1 Report
Enclosed please kindly find the comments and suggestions. Thank you.

Thank you for the invitation.
Reviewer 2 Report
R
Author Response
Thanks to the reviewer for the comment.
Reviewer 3 Report
Nil
Nil
Author Response
Thanks to the reviewer for the comment.